# Social Media Analytics and Metrics for Improving Users Engagement

**Ioannis C. Drivas \***, **Dimitrios Kouis**, **Daphne Kyriaki-Manessi and Fani Giannakopoulou**

Information Management Research Lab, Department of Archival, Library and Information Studies University of West Attica, 12243 Egaleo, Greece; dkouis@uniwa.gr (D.K.); dkmanessi@uniwa.gr (D.K.-M.); fgiannakopoulou@uniwa.gr (F.G.)
**\*** Correspondence: idrivas@uniwa.gr

**Abstract:** Social media platforms can be used as a tool to expand awareness and the consideration of cultural heritage organizations and their activities in the digital world. These platforms produce daily behavioral analytical data that could be exploited by the administrators of libraries, archives and museums (LAMs) to improve users' engagement with the provided published content. There are multiple papers regarding social media utilization for improving LAMs' visibility of their activities on the Web. Nevertheless, there are no prior efforts to support social media analytics to improve users' engagement with the content that LAMs post to social network platforms. In this paper, we propose a data-driven methodology that is capable of (a) providing a reliable assessment schema regarding LAMs Facebook performance page that involves several variables, (b) examining a more extended set of LAMs social media pages compared to other prior investigations with limited samples as case studies, and (c) understanding which are the administrators' actions that increase the engagement of users. The results of this study constitute a solid stepping-stone both for practitioners and researchers, as the proposed methods rely on data-driven approaches for expanding the visibility of LAMs services on the Social Web.

**Keywords:** social media platforms; Facebook; social media networks; social media data; analytics; metrics; libraries; archives; museums; users' engagement

## 1. Introduction

Social Web or Web 2.0 refers to a set of social relations that interconnects people, helping them to communicate through the World Wide Web [1]. Over the last decade, Social Media Platforms (SMPs) such as Facebook, Twitter, YouTube and Instagram have become vital neurons of the social web ecosystem, becoming increasingly popular for individuals and various organizations [2]. Social media platforms allow people to communicate, contribute to content creation as well as engage and interact with the published content. When it comes to the usage of SMPs in libraries, archives and museums (LAMs), the purpose is not articulated just to cover users' information needs by LAMs administrators. Taking one step further, SMPs constitute a powerful tool to improve brand awareness and consideration of LAMs [3]. They also work as a fast-spreading vehicle to promote services and, thus, expand their web visibility and, most importantly, allow stakeholders to engage with the published content of such organizations through reactions, shares, and comments [4].

It is common knowledge that several studies focus on the SMPs' importance in LAMs context [5–9]. On the one hand, this fact positively pushes the limits to established well-organized research frameworks with the purpose to understand users' interaction in SMPs of specific LAMs as cases [3,10]. On the other hand, wider contexts of research efforts through funding projects in the European Union are examined, aiming to establish policies for efficient social media data encapsulation within the LAMs sector (e.g., PLUGGY: Pluggable Social Platform for Heritage Awareness and Participation—https://cordis.europa.eu/

project/id/726765 (accessed on 11 March 2022), UNCHARTED: Understanding, Capturing and Fostering the Societal Value of Culture—https://cordis.europa.eu/project/id/870793 (accessed on 11 March 2022)). LAMs administrators need to aptly examine users' engagement with the published content through SMPs and understand what types of content improve engagement levels [11]. Social media analytics (SMAs) and metrics can measure users' engagement with the published content of LAMs by using different platforms [12]. Social media analytics and metrics are set under the strategic framework of integrating information technology tools to harvest, pre-process, analyze and summarize the produced social media data to accomplish specific organizational goals.

Nevertheless, research efforts around the utilization of SMAs and metrics for LAMs indicate several difficulties. First, after examining their empirical findings, most of the studies indicate LAM staff difficulties, realizing the value that SMAs produce in order to utilize them in LAMs context. Second, several studies examine SMAs in specific organizational contexts as cases, without including multiple different LAMs within their data sample. This fact limits the holistic overview and knowledge that administrators should have about users' engagement with the provided published content of LAMs. Moreover, very few approaches proposed an assessment schema to measure users' engagement within LAMs context. However, these approaches did not consider the issue of the statistical reliability that could allow administrators to replicate the proposed methodologies to measure users' engagement in SMPs of the LAMs they manage. Beyond these drawbacks, it is also noted that, to the best of our knowledge, there are no published research efforts that indicate which of the administrators' actions (such as types of posts or post frequency) impact users' engagement (reactions, comments and shares) with the published content of LAMs through SMPs.

Against this background, in this study, we propose a three-stage data-driven methodology to capture, analyze and interpret SMAs to identify the administrators' actions that increase users' engagement with the published content through SMPs. More specifically, at the initial stage, social media data coming from Facebook for 220 libraries, archives and museums worldwide were collected. Subsequently, the study verifies the statistical reliability and the internal consistency of the collected SMAs, aiming to provide potential researchers and administrators with an instrument for measuring users' engagement in Facebook. At the third stage, the paper proceeds into the development of predictive regression models, further examining which are the administrators' actions that bring upon greater users' engagement with the published content of the examined LAMs.

To this end, the study is organized into five sections. First, an overall conceptualization regarding social media analytics and users' engagement research topics is provided. This helps the readers of the study to understand the importance of these topics and how they could be utilized to develop new social media strategies or optimize existing ones. Thereafter, a scientometrics analysis is provided, highlighting the importance of SMPs and SMAs within the research context of LAMs. Hereupon, the related prior research efforts are presented while also designating the research gaps. In Section 3, the proposed data-driven methodology is described in detail. After that, in Section 4, the results are presented. Lastly, Section 5 is devoted to discussing the results, the study's practical contribution and the future implications.

## 2. Related Background

### 2.1. Conceptualizing Social Media Analytics and Metrics—An Overall Point of View

As big data analytics characterized more scientifically matured than the previous decade's beginnings [13], the vision expressed by Kaplan and Haenlein [14] for efficient social media analytics methods starts to flourish. Social media platforms produce voluminus and volatile data that could be monitored, measured and analyzed by the organizations' administrators for improving the online social presence and visibility of their services, products and activities. To achieve this goal, social media analytics techniques and strategies are used. From the academic perspective, according to [15], SMAs entail "the process

of developing and evaluating informatics tools to measure the activities of users within social media platforms. Social data are derived from conversations, users' engagement with posts, sentiment, influence, and other attributes that can be collected, monitored, analyzed and visualized". According to Zeng et al. [16], SMAs are defined as "an interdisciplinary research field that aims to combine, extend and adapt methods to analyze social media data". From practitioners' perspective, SMA is an evolving business topic that encapsulates and analyzes online conversation (industry, competitive, prospect and user/customer) and social activity articulated by organizations through social network platforms [17]. More specifically, Awareness Inc. states that "Social analytics enable organizations to act on the derived intelligence for business results, improving brand awareness and reputation, marketing and sales effectiveness, and customer satisfaction and advocacy".

By conducting a detailed scientometrics analysis, Misirlis and Vlachopoulou [18] mapped that most of the papers around the topic of SMAs and metrics related to users' engagement and behaviour (38.4% of the total papers examined), while the rest focus on other disciplines (awareness and branding: 13.4%; predictive marketing research: 9.6%; social capital, value and ROI: 15.4%). The increased interest in the sub-topic of users' engagement and behaviour in SMPs could be justified by prior research approaches. Measuring social media users' engagement and reactions provides organisations the capabilities to predict users' loyalty to products, services and activities [19]. It also allows understanding what type of content users are more engaged in [20], which can potentially improve their trust and affinity with the organizations' online presence [21].

Social media engagement constitutes a multidimensional and polysemic research field [22]. By examining the systematic literature review of Trunfio and Rossi, research around social media engagement is framed into four areas. Research around users' behaviour, platforms and tools for harvesting social media engagement data; the derived metrics; and theoretical efforts for greater conceptualization of the topic. Hallock et al. [23] proposed a theoretical definition of social media engagement, defined as "what occurs as a user builds relationships with other users and brands. It is more than merely liking, commenting or posting within a social network. Instead, it reveals a longer-term relationship among users". From a more practical point of view, Le [24] described social media engagement as the capability of measuring users' online behaviour via the so-called engagement metrics of actions. These include the number of users, click-through rates, page views, content likes, comments or reactions depending on the SMP that the organizations use. Therefore, based on the practical point of view, analysts and administrators of organizations should focus more on the behavioural interactions associated with likes, shares and comments when trying to encapsulate quantitatively social media engagement per platform [25].

Libraries, archives and museums—as organizations that foster and expand users' interactions with the cultural content on the Social Web—could not be an exception in terms of SMAs, metrics and the methods used to estimate users' engagement. In Section 2.2, a scientometrics analysis reveals the increased research activity around the topic. Subsequently, in Section 2.3, the related research efforts around SMAs and metrics utilization for measuring users' engagement in LAMs organizations are unfolded.

*2.2. Importance of Social Media Platforms and Analytics for LAMs—A Scientometrics Analysis*

While SMPs started their initial acquaintances with web users almost 18 years ago, mainly through the Facebook platform, research efforts regarding their utilization within the LAMs context have not started before 2009.

More specifically, by accessing the Scopus citation index with the following query string in the advanced search field "TITLE ("social media" OR "social media analytics" OR "social media metrics") AND TITLE ("libraries" OR "archives" OR "museums") AND PUBYEAR < 2022", 310 published documents were returned from 2009 up to 2021 (Figure 1). Based on the extracted results, it is noted that there has been a tremendous increase in document publication activity over the last four years (2018: 32 documents; 2021:

42 documents). Without a doubt, this entails a recent overall increase in related researchers' efforts to use SMPs and SMA for improving LAMs services on the Web. A fact that designates the importance for LAMs and how the related research activity pays attention to this specific topic.

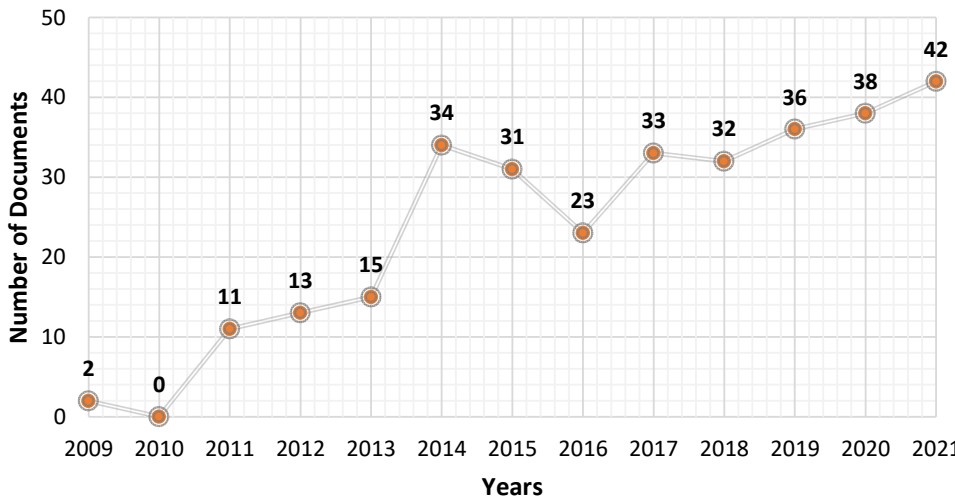

**Figure 1.** Publication activity around social media and LAMs between 2009 and 2021. The horizontal axis indicates the years, while the vertical axis indicates the number of documents per year.

Furthermore, the following figure (Figure 2) depicts the publication type for the documents retrieved through the Scopus citation index. Most of the documents have been published in journals as articles (220 documents), while books and book chapters come in second place (42 documents).

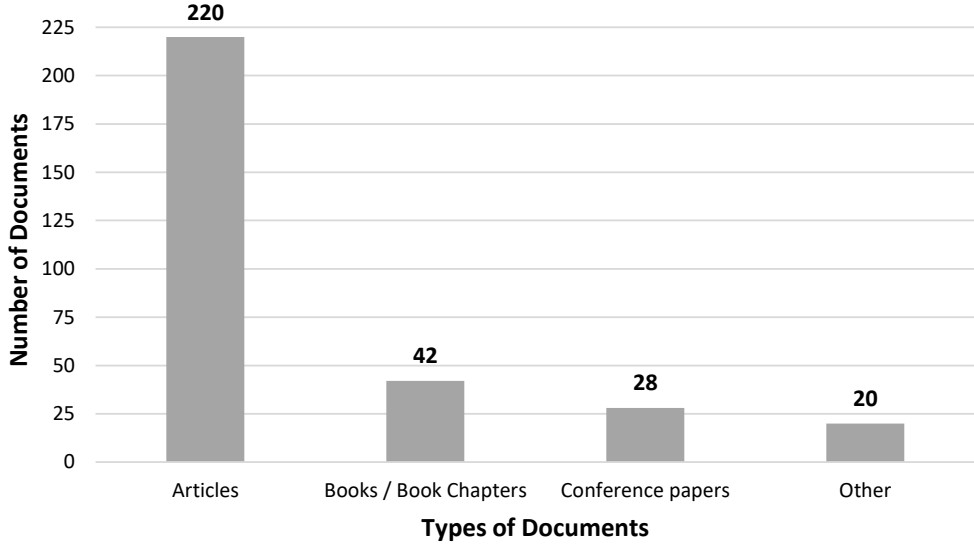

**Figure 2.** Document types extracted after conducting the search query in the Scopus search engine.

Further analyzing Scopus results, in Figure 3, we present the research activity per country around the topics of SMPs and SMAs, and their utilization on LAMs. We considered the authors' affiliation origin to identify the country per document. A world map is illustrated on the left side of the figure. The more publications there are per country, the more intense the color tone on the map. On the right side, the bar chart presents the number of documents per country.

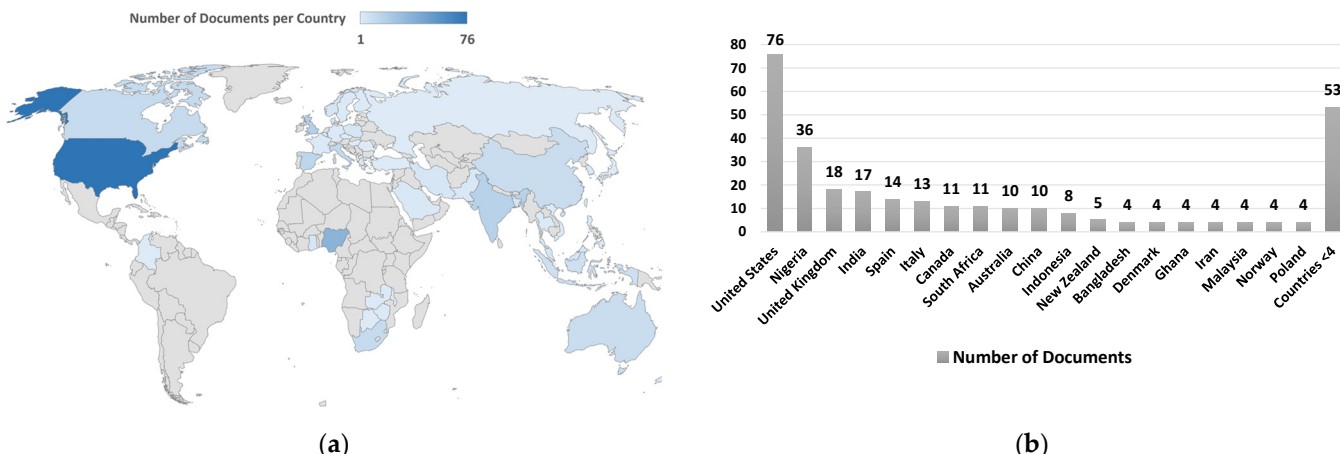

**Figure 3.** Research activity per country around SMPs/SMAs and their utilization on LAMs: (**a**) world map; (**b**) number of documents per country.

As it can be seen, the researchers from the United States published most of the documents on the SMPs and SMAs topics in LAMs (76 documents). What is noteworthy is the publication activity in Nigeria, reaching up to 36 published documents. Lastly, there is sporadic publication activity in other countries, reaching 53 published documents with 3 or fewer documents per country.

Based on the vigorous research activity, the prior related efforts and research gaps are discussed in the next section.

### 2.3. Prior Efforts and Research Gaps

In the era of Big Data and the robust and voluminous social data production, Cervone [26] focused on LAMs social media analytics exploitation and highlighted why it is important to understand the relation between users' engagement with the published content by providing five fundamental reasons: (i) tracking the growth of presence, (ii) understanding how content performs and resonates among users, (iii) understanding audience characteristics, (iv) observing upcoming trends, and lastly, (v) tracking progression towards pre-defined key point indicators.

Within this context, Mensah et al. [10] performed an in-depth investigation about social media utilization in academic libraries' context. They developed a quantitative survey to examine library staff's willingness to utilize SMPs to promote library services. However, although their results indicate that staff agreed to develop social media strategy as the most appropriate tool to increase users' awareness about libraries' services, they did not possess the expected skills to understand users' engagement with the provided content. The same conclusions were also stated in a prior investigation by Jones and Harvey and published in the Journal of Academic Librarianship [27]. Cheng and colleagues [3] conducted a comparative study between the users' and library staff perspectives regarding the effectiveness of Facebook as a marketing tool in the Hong Kong University libraries. Their findings indicated no satisfactory reactions and engagement from users regarding the published library content. Their research also proved that the interactions of the current users affect the future visitors' attitudes toward accepting the library's Facebook page as a social media tool to promote content and services.

Understanding users' engagement with the provided content through the utilization of SMAs constitutes a well-informed approach to improve and support the LAMs social media strategy. For example, Pozas [28] exploited the produced SMAs from the National Library of Spain. A strong relationship between social media campaigns and a significant increase in digital collections usage was discovered. The research of Pozas [28] relied on a prior proposed framework by González-Fernández-Villavicencio [29], who suggested six categories of social media metrics: (i) Reach (popularity, size and visibility), (ii) Engagement

(comments, shares, views, downloads, etc.), (iii) Loyalty (website traffic coming from social media), (iv) Influence (users' brand perception: mentions, sentiment, reputation index), (v) Activity frequency, (number of posts, uploads, etc.) and (vi) Conversion (number of downloads of digital collections, downloads of tutorials, number of loans, etc.).

Another study utilizes SMAs for understanding what type of Twitter content "inspires" museum users. The primary purpose was to adopt similar content development and distribution approaches for future users [30]. The proposed method for encapsulating users' inspiration through Twitter data resulted in up to 67% f-measure score of effective retrievals. The results of [30] also support the assumption that SMAs could be used to understand users' engagement with content and, hence, produce material according to their preferences. In a similar vein, Ref. [31] proposed a framework to utilize Twitter analytics to understand the most appropriate metrics that could be used to quantify users' engagement with the Tate Modern Museum of Art content. The appropriate metrics were identified by using the Balance Scorecard theory: the Number of tweets sent by Tate, the Impressions, the Retweets, the Favourites, the Replies and the Other Interactions.

Another effort that utilizes the SMAs provided by Instagram to increase awareness of specific collections in the museum sector is the Metropolitan Museum of Art [32]. The author suggests several data-driven actions to attract more followers based on five metrics categories which collect information about users' profiles and engagement with the provided content, which includes Users actions (profile views and website clicks), Discovery (reach of posts and impressions), Followers (demographics, time that followers are most active and top locations), Media and Promotions (metrics related to the performance of the published posts) and Stories (metrics related with users' engagement with stories such as taps forward, taps back, replies, swipes away, etc.).

Social media analytics contribution for promoting LAMs services to relevant stakeholders has also been stated by Boulton [33]. The author examined the utilization of SMPs for the institutional repository of Griffith University and pointed out their contribution through two pillars: first, to measure users' interaction with the provided content and estimate the success of the deployed strategies for promoting repositories services; secondly, how management staff could use SMAs to overhaul communication and relationships among different teams within the university (librarians, marketing team, students, researchers, etc.). Furthermore, in the archival sector, Magier [34] investigated how SMAs could be used in the State Archives in Siedlce to inform users and keep the same level of engagement with them during the covid era.

Without a doubt, all the studies mentioned above highlight the contribution of SMPs and the derived SMAs to understand users' engagement with the published content and set guidelines for increasing the awareness of the services provided by LAMs. Most of the studies proceed into the examination of a particular case of LAM and how it utilizes social media platforms and analytics to accomplish specific organizational goals [28,30–34]. Other studies used structured questionnaires to investigate users' interaction with the LAMs' social media platforms and measure their engagement with the published content [3,10].

Nevertheless, very few studies [3,10] involved a large number of LAMs cases and their SMAs in providing a holistic approach and consequently a generalization framework on how the latter could be utilized to increase users' engagement and expand the awareness of LAMs organizations.

In our paper, we intend to provide concrete evidence that the utilization of social media analytics and metrics could be a reliable alternative for understanding users' engagement with the published content in an SMP by a LAM organization. Our efforts focus on providing an additional framework for measuring users' engagement through SMAs while simultaneously collecting a more representative number of LAMs as use cases. This framework could support and complement previously established efforts that collected users' engagement data within an SMP by using a questionnaire as a research instrument. Moreover, assessing the proposed framework's validity, reliability and internal consistency could result in future studies adopting it in the LAMs and other domains, as the same level

of reliability is expected [35] (p. 32) [36]. The following table (Table 1) summarizes the research context issues and the contributions of the present study.

**Table 1.** Research Context Issues and Contributions of the present study.

| Research Context Issues | Contributions |
|---|---|
| There is a need to improve the social media skills of staff for understanding users' engagement with the uploaded content [3–10]. | Understanding social media analytics and metrics and the possible intercorrelations between them will improve staff skills in providing content that results in higher levels of users' engagement. |
| The majority of the current studies proceed into individual examinations of how a LAM utilizes SMPs to understand and measure users' engagement with the published content [28,30–34]. | Further research is needed to provide a holistic approach and consequently a generalization framework on how SMAs could be utilized to increase users' engagement and expand the awareness of LAMs organizations. This could also work as a benchmarking process for the administrators of the LAMs. |
| Lack of a SMAs methodological framework that exhibits validity, reliability and internal consistency in terms of the included variables that measure LAMs users' engagement with the published content [35,36]. | Suggest an assessment schema that expresses statistical reliability in its nature. This schema will quantitatively measure users' engagement within an SMP of a LAM. |

In the following section, the research methodology is unfolded to cover the research gaps that have been discussed previously.

## 3. Methodology

### 3.1. Data Collection and Sample

Based on prior research efforts for collecting information about LAMs and their presence on the Web [37], a dataset consisting of 341 domains was developed. Data about the available social media pages were gathered from each LAM's website. In most cases, this information was located on the footer of the websites or on the contact pages as social media icons, including a link that redirected users to the social media page. The social media pages from various SMPs for 220 LAMs organizations were collected at the end of this process. Several cases were excluded from the initial dataset as they did not have a social media page or had no activity for the last six months.

Fanpage Karma (https://www.fanpagekarma.com/ (accessed on 17 March 2022)) was used to gather data from the social media pages in our case, as in similar research efforts [38,39]. Fanpage Karma tool provides several metrics for each SMP, including Facebook, Twitter, YouTube, Instagram and LinkedIn. We extracted metrics for each SMP regarding users' behaviour and administrators' actions (such as types of posts or posts' frequency) in a time range of 30 days for all the 220 LAMs organizations.

Only the Facebook social media platform and its retrieved metrics were finally chosen to be analyzed in this paper. Based on the literature review, there are multiple papers over the last ten years that highlight the contribution of Facebook as an SMP for LAMs. In libraries context, Cassidy et al. [40] reported that more than 70% of students selected Facebook as a communication tool to provide library services in the Sam Houston State University library. In a similar approach, Okoroma [41] indicated that Facebook constitutes the preferable SMP for reference services to students, among others. Mensah et al. [10] study articulated up to 81% Facebook preference for staff and 56% for library patrons.

Furthermore, in the museums' context, Fissi et al. [42] and Camarero et al. [43] highlighted Facebook's prominence among other SMP usage for cultural heritage institutions through their research. Mukwevho and Ngoepe [44] identified Facebook as the most common SMP to promote archival material to society in the archival organizations' context. In the same way, Magier's data analysis [34] proved that Facebook activity increased during the COVID-19 pandemic among other social networks, both among the staff and patrons at the State Archives in Siedlce. Similar approaches have been followed by Tkacová et al. [45].

Another reason for choosing Facebook SMP is based on preliminary research efforts regarding SMP and their contribution to website traffic in LAMs. Through the Similar

Web API, the leading social networks that drove traffic to each one of the cases in our dataset were identified, confirming the popularity of Facebook (for more details, see in the Data Availability Statement the Dataset Traffic Acquisition to LAMs Websites). Specifically, the average incoming website traffic for a month monitoring period from social networks was approximately 2.68% (social traffic). The analysis of the results indicated that Facebook was responsible for 46.1% of social traffic. The following graph (Figure 4) depicts the SMPs responsible for creating incoming traffic and their percentage share. Therefore, both the findings of the related literature review and the results of our preliminary data harvesting process support choosing Facebook as the most appropriate SMP compared to others.

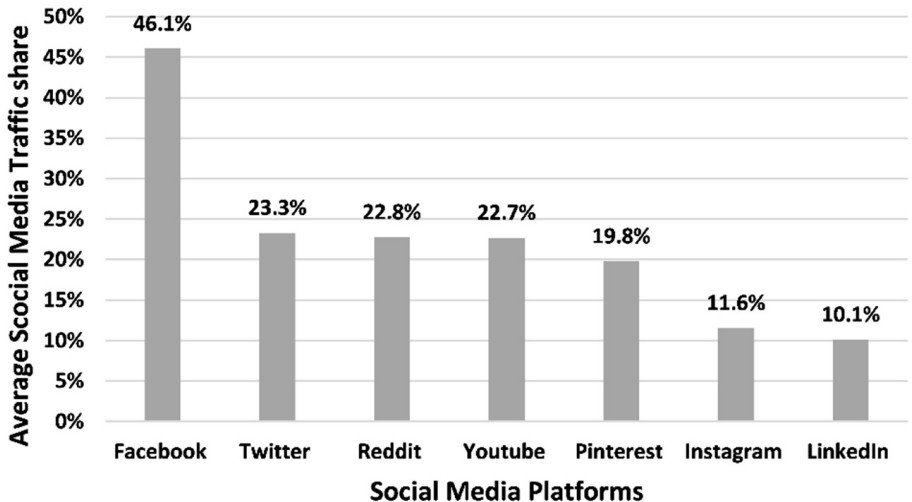

**Figure 4.** Average incoming social media traffic share per SMP.

The following table (Table 2) explains Facebook metrics, allowing readers to understand the possible causal relationship.

**Table 2.** Description of Facebook metrics involved in this study.

| Metric Name | Metric Description |
|---|---|
| Number of Posts | Number of posts that have been published in a specific period. |
| Link posts | It is the number of posts in URL format that have been published in a specific period. |
| Picture posts | It is the number of posts in picture format that have been published in a specific period. |
| Video Posts | Number of posts in video format that have been published in a specific period. |
| Comments per post | The average number of comments on posts in a specific period. |
| Number of reactions | The total number of (like, love, haha, thankful, wow, sad, angry) on posts that have been published in a specific period. |
| Reactions per Post | The average number of reactions on posts that have been published in a specific period. |
| Number of Comments (total) | It refers to the total number of comments on posts. This includes answers to these comments that have been published in a specific period. |
| Total Reactions, Comments, Shares | It expresses the number of reactions of any type (like, love, haha, thankful, wow, sad and angry), comments and shares on posts that the LAM organization has published in a specific period. |

For the 220 LAMs cases, the Facebook metrics were calculated for a 30-day harvesting period. In the following table (Table 3), a sample of the overall dataset is presented. The first column depicts the name of LAM, while the rest provide data for the metrics. The sample presented allows readers to understand the upcoming steps of the proposed methodology, namely the validity and reliability of the metrics and the development of predictive regression models.

**Table 3.** Sample of the dataset regarding a random selection of three different LAMs.

| Name of LAM | Number of Posts | Link Posts | Picture Posts | Video Posts | Comments per Post | Number of Reactions | Reactions per Post | Number of Comments (Total) | Total Reactions, Comments, Shares |
|---|---|---|---|---|---|---|---|---|---|
| Denver Art Museum | 29 | 3 | 17 | 3 | 2.52 | 1799 | 78.21 | 58 | 2105 |
| National Library of Spain | 19 | 1 | 9 | 5 | 7.25 | 3940 | 246.25 | 116 | 5512 |
| National Archives of Georgia | 34 | 3 | 22 | 6 | 1.22 | 1847 | 59.58 | 38 | 2242 |

### 3.2. Validity and Reliability Assessment

After harvesting the social media analytics data for Facebook, a statistical analysis was performed to validate their reliability and consistency. Before the reliability analysis, a preliminary examination of the first 25% of the dataset was conducted. The aim was to ensure that the retrieved dataset expressed normality [46], which was necessary to construct a valid predictive model through linear regressions [47]. We used Shapiro–Wilk as the most potent normality indicator among Kolmogorov–Smirnov, Lilliefors and Anderson–Darling tests [48].

In addition, by using the descriptive statistics approach, we also measured skewness to understand, in a practical way, the initial situation of the Facebook page performance of a LAM. Skewness measures the tendency of a variable between the minimum and the maximum values contained [49]. Negative skewness indicates that most of the variables' values tend to be closer to the maximum value. In contrast, a positive value indicates that most values tend to be closer to the minimum value. This helps in understanding the overall tendency of the proposed Facebook metrics and if each of the examined LAM tends to be closer to the minimum or maximum values.

Facebook metrics were divided into two factors. The first factor contains the metrics that indicate administrators' actions on a LAM Facebook page: Number of Posts, Link posts, Picture posts and Video posts. The second factor contains the metrics that express followers' engagement on a LAM Facebook page: Comments per post, Number of reactions, Reactions per post, Number of comments (total) and Total reactions, comments and shares. This categorization will allow administrators to understand which of their actions impact, at a lower/higher level, the followers' engagement with the posts. In other words, some of the administrators' actions may impact specific metrics of engagement more than others. For example, posting pictures probably resulted in more total reactions, comments and shares by followers than the link posts or video posts. An exploratory factor analysis (EFA) was also conducted to ensure the statistical significance of the two factors of metrics. By using EFA, Kaiser–Meyer–Olkin (KMO), Bartlett's test of Sphericity and $\chi^2$ tests were performed to test the goodness of fit of each variable relative to the two proposed factors [50].

One step further, the study aims to provide a reliable assessment model regarding LAMs Facebook performance page through the involvement of several variables that express validity, reliability and cohesion. As stated before, if model reliability tests are successful, then there is a high probability that this could also apply in other domains [35] (p. 32). On this basis, to testify to the reliability of the proposed model, we deploy four different statistical tests: McDonald's $\omega$, Cronbach's $a$, Guttman's $\lambda$-2 and $\lambda$-6, respectively. McDonald's $\omega$ estimates the strength association among the involved variables within a factor [51]. The greater the association among the variables, the closer the value to 1, while the lower the association, the closer the value to 0. Cronbach's $\alpha$ estimates the acceptance level of the two proposed factors, while Guttman's $\lambda$-2 reinforces the Cronbach's results by measuring the variance trustworthiness among the selected variables in each factor [52,53].

In the effort to construct linear predictive models that estimate the potential impact of administrators' actions on users' engagement, Guttman's $\lambda$-6 test was also conducted. The latter calculates the variance in each variable involved within the linear regression proposed models [54]. Lastly, by deploying Variance Inflation Factor (VIF), it is ensured

that the proposed constructs (users' engagement and administrators' actions) do not face multicollinearity issues [55].

### 3.3. Predictive Regression Models

After categorizing metrics into two factors, namely, administrators' actions and users' engagement, we developed linear predictive regression models to understand the cause-and-effect relationship among the involved metrics. More specifically, it will be helpful to understand the predicted value of change in users' engagement metrics if each of the administrators' actions metrics increases by one unit. For example, if picture posts are increased by one unit, there will be an increase in the metric of total reactions, shares and comments. The results of the predictive regression models will help LAM administrators identify which of their post strategies bring higher levels of users' engagement with content.

In the following figure (Figure 5), we present the proposed data-driven methodology, helping the readers of the study wrap up all the actions that have been made in each stage, while in the next section, the results are presented.

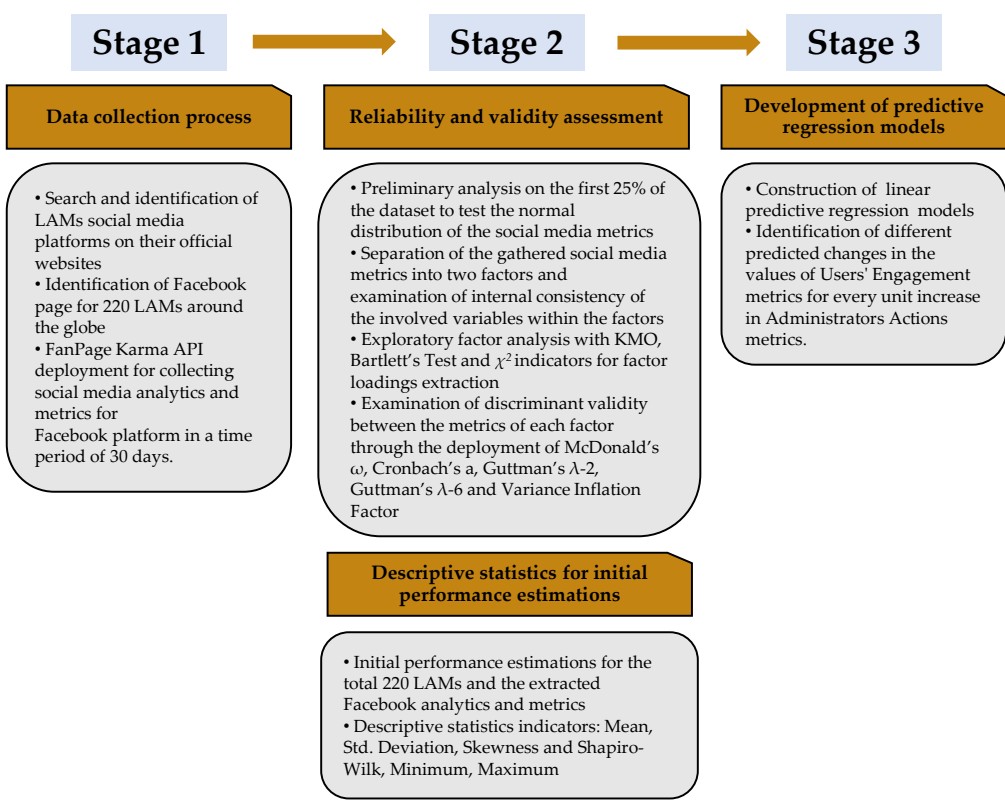

**Figure 5.** Three stages of the proposed data-driven methodology in a sequential way.

## 4. Results

### 4.1. Validation of the Proposed Factors

Following the proposed methodology, the exploratory factor analysis (EFA) and reliability analysis results are presented first. Table 4 reports the values for each one of the factors: the variable loadings for each metric and the goodness of fit tests (KMO, Bartlett's test of Sphericity and $\chi^2$). In addition, KMO extracted higher values for the two factors (0.634 for Administrators Actions and 0.708 for Users Engagement) than the recommended tolerance of exclusion (<0.50).

**Table 4.** Exploratory factor analysis results and loading per factor.

| Administrators Actions | | Users Engagement | |
|---|---|---|---|
| *Variables* | *Variable Loading* | *Variable* | *Variable Loading* |
| Number of posts | 0.767 | Comments per post | 0.706 |
| Link posts | 0.519 | Number of reactions | 0.727 |
| Picture posts | 0.667 | Reactions per post | 0.690 |
| Video posts | 0.624 | Number of comments (total) | 0.655 |
| | | Total reactions, comments, shares | 0.751 |
| *0.634 \* | < 0.001 \*\* | < 0.001 \*\*\** | | *0.708 \* | < 0.001 \*\* | < 0.001 \*\*\** | |

\* KMO, \*\* *p*-value of Bartlett's test of Sphericity, \*\*\* *p*-value of $\chi^2$.

The following table (Table 5) presents the proposed factors' reliability and internal consistency results. As observed, Administrators Actions designated high reliability with values ranging from 0.748 (Cronbach's $\alpha$) up to 0.967 (McDonald's $\omega$). In a similar vein, Users Engagement indicated sufficient reliability starting from 0.648 (Cronbach's $\alpha$) up to 0.934 (Guttman's $\lambda$-6). Moreover, the Administrators actions' VIF values and Users' engagement were below the tolerance of 3.3, as Diamantopoulos and Siguaw [56] suggest; thus, no multicollinearity issue was observed.

**Table 5.** Reliability analysis and internal consistency of the two factors.

| Factors | McDonald's $\omega$ | Cronbach's $\alpha$ | Guttman's $\lambda$-2 | Guttman's $\lambda$-6 |
|---|---|---|---|---|
| Administrators Actions | 0.967 | 0.748 | 0.847 | 0.917 |
| Followers Engagement | 0.915 | 0.648 | 0.889 | 0.934 |

Based on the above results, there is a high probability that the proposed factors and their metrics can extract similar reliability and internal consistency in a different dataset of Facebook pages of other domains rather than LAMs [35,36]. If future research approaches adopt the proposed methodology to measure Facebook pages' performance of other organizations apart from LAMs, it is expected to have similar reliability and internal consistency results.

The second stage of the proposed methodology presents the results of the descriptive statistics for the two factors and their metrics.

*4.2. Descriptive Data Summarization for Initial Performance Estimations*

The upcoming tables (Tables 6 and 7) contain the descriptive results of the two factors and their social media metrics. The following descriptive statistics represent the administrative actions (Table 6) and users' engagement (Table 7) for a 30 days period.

**Table 6.** Descriptive statistics results for the variables included within Administrators Actions factor (30 days period).

| | **Number of Posts** | **Link-Posts** | **Picture Posts** | **Video-Posts** |
|---|---|---|---|---|
| Mean | 27.591 | 3.695 | 19.447 | 4.448 |
| Std. Deviation | 23.498 | 6.069 | 17.448 | 8.498 |
| Skewness | 2.304 | 4.882 | 2.206 | 4.814 |
| Shapiro-Wilk | 0.811 | 0.563 | 0.816 | 0.539 |
| Minimum | 1.000 | 1.000 | 1.000 | 1.000 |
| Maximum | 158.000 | 56.000 | 114.000 | 75.000 |
| *N = 220 | p-value of Shapiro–Wilk $\leq$ 0.001* | | | | |

**Table 7.** Descriptive statistics results for the variables included within the Users Engagement factor.

|  | Comments per Post | Number of Reactions | Reactions per Post | Number of Comments (Total) | Total Reactions, Comments, Shares |
|---|---|---|---|---|---|
| Mean | 3.562 | 3148.467 | 101.424 | 121.029 | 3890.619 |
| Std. Deviation | 6.278 | 5706.804 | 159.020 | 262.374 | 7155.607 |
| Skewness | 3.506 | 3.127 | 2.964 | 4.143 | 3.147 |
| Shapiro-Wilk | 0.574 | 0.576 | 0.613 | 0.487 | 0.565 |
| Minimum | 1.000 | 1.000 | 1.000 | 1.000 | 1.000 |
| Maximum | 44.429 | 35,424.000 | 871.000 | 2109.000 | 40,991.000 |

*N = 220 | p-value of Shapiro–Wilk ≤ 0.001*

In terms of the administrators' actions metrics, all of them extracted significant Shapiro–Wilk $p$-values (<0.001), indicating statistical normality. Moreover, all metrics extracted positive skewness, resulting in their values tending more to the minimum than the maximum points. Descriptive statistics indicate that Picture Posts are the most frequent type of publication (Mean: 19.447) for LAMs Facebook pages compared to Link Posts (Mean: 3.695) and Video Posts (Mean: 4.448). It is also noted that the mean value of the Number of posts reaches up to 27.591. This practically means that the examined LAMs post on Facebook almost every day at least one type of post (27.59/30 = 0.91 posts per day).

The Users Engagement metrics are presented in the following table (Table 7). Regarding their normal distribution, all the metrics resulted in sufficient Shapiro–Wilk values ranging from 0.487 (Number of Comments (total)) up to 0.613 (Reactions per post) with statistically significant $p$ values at <0.001.

In the same line with Administrators' Actions metrics, all metrics in this factor extracted positive skewness values ranging from 2.964 up to 4.143, indicating their tendency closely to minimum points. In addition, as shown in Table 7, the examined LAMs receive on average 3.56 comments per post and a total mean value of 121.02 comments within the period of 30 days. Finally, the Total Reactions, Comments and Shares of the examined 220 LAMs extracted a mean value of 3890.61.

*4.3. Predictive Regressions Results*

Tables 8–10 present the regression equation outputs. The scope is to indicate the potential predicted change in the metrics within the Users Engagement factor if specific administrators' actions are performed. In most cases, the extracted regression predictive models indicated high statistical significance with $p < 0.001$. F value is also included. The results support the assumption that the proposed predictive models can reject the null hypothesis, which is that the regression coefficients are equal to zero values; thus, the model lags behind predictive discriminant capability [57]. The only exception appears for the Link Posts metric. The results confirm the statistically non-significant interdependence and correlation between Link Posts and the Users' Engagement metrics (results for Link Posts metrics are depicted in Tables 8–10 with a strikethrough line).

**Table 8.** Regression equation output of the Number of Total Reactions, Comments and Shares and their potential predicted change in each of the proposed Administrator Actions.

| Variable | Coefficient | $R^2$ | F | *p*-Value |
|---|---|---|---|---|
| Constant (Number of Total Reactions, Comments, Shares) <br> *Number of Posts* | 785.88 <br> 122.02 | 0.154 | 38.44 | <0.001 |
| Constant <br> *Link Posts* | ~~4097.08~~ <br> ~~−12.19~~ | ~~0.000~~ | ~~0.007~~ | ~~<0.931~~ |
| Constant <br> *Picture posts* | 806.75 <br> 168.01 | 0.155 | 38.25 | <0.001 |
| Constant <br> *Video Posts* | 2802.88 <br> 454.38 | 0.122 | 19.16 | <0.001 |

**Table 9.** Regression equation output of the Number of Comments (total) and their potential predicted change in each of the proposed Administrators' Actions.

| Variable | Coefficient | $R^2$ | F | *p*-value |
|---|---|---|---|---|
| Constant (Number of Comments Total) <br> *Number of Posts* | 9.723 <br> 4.46 | 0.171 | 35.47 | <0.001 |
| Constant <br> *Link Posts* | ~~133.23~~ <br> ~~−2.28~~ | ~~0.002~~ | ~~0.227~~ | ~~<0.635~~ |
| Constant <br> *Picture posts* | 19.548 <br> 6.55 | 0.108 | 21.61 | <0.001 |
| Constant <br> *Video Posts* | 70.64 <br> 16.68 | 0.102 | 22.18 | <0.001 |

**Table 10.** Regression equation output of the Number of Reactions and their potential predicted change in each proposed Administrators Action.

| Variable | Coefficient | $R^2$ | F | *p*-Value |
|---|---|---|---|---|
| Constant (Number of Reactions) <br> *Number of Posts* | 78.87 <br> 127.26 | 0.163 | 40.66 | <0.001 |
| Constant <br> *Link Posts* | ~~3232.98~~ <br> ~~−4.80~~ | ~~0.000~~ | ~~0.002~~ | ~~<0.935~~ |
| Constant <br> *Picture posts* | 170.83 <br> 164.79 | 0.164 | 38.96 | <0.001 |
| Constant <br> *Video Posts* | 2490.82 <br> 322.38 | 0.107 | 15.78 | <0.001 |

In Table 8, the potential predicted changes of the metric Number of Total Reactions, Comments and Shares are presented if Administrators Actions metrics are increased by one. To begin with, a high statistical significance was observed with $p < 0.001$ and $R^2$ of 0.154 between the Number of Total Reactions, Comments, Posts and the Number of Posts. More specifically, for each new post on the Facebook page of the examined LAMs, the Number of Total Reactions, Comments and Shares is expected to increase by 122.02. Picture posts ($R^2$ 0.155 and $p < 0.001$) resulted in an increase in the Number of Total Reactions, Comments and Shares by 168.01. The highest engagement is observed for video posts. A significant regression equation was observed with $p < 0.001$ and $R^2$ of 0.122 between the Number of Total Reactions, Comments and Shares and Video Posts. In more detail, for every new video post on the Facebook page of the examined LAMs, the Number of Total Reactions, Comments and Shares could be increased by up to 454.38.

Continuing the presentation of the regression predictive models and their results, Table 9 depicts the potential predicted change of the metric Number of Comments (total) if Administrators Actions metrics are increased by one. More specifically, for each new post on the Facebook page of the examined LAMs, the Number of Comments (total) is expected to increase by 4.46. Picture Posts indicated that they could result in a higher number of comments, as for each new Picture Post, the Number of Comments increases by 6.55 ($R^2$ 0.108 and $p < 0.001$). In the same line with the results of Table 8, Video Posts extracted the highest predicted change. More specifically, for each new video post, the Number of Comments within a period of 30 days could be increased by up to 16.68 ($R^2$ 0.102 and $p < 0.001$).

Lastly, in Table 10, the potential predicted change of the dependent variable Number of Reactions (like, love, haha, thankful, wow, sad and angry) is presented. In line with previous results (Tables 8 and 9), Video Posts extracted the highest impact on the Number of Posts metric. More specifically, for each new Video Post, the Number of Reactions could be increased by 322.38 ($R^2$ 0.107 and $p < 0.001$). Moreover, for every new Picture Post published on the LAMs Facebook page, the Number of Reactions could be increased by 164.79 ($R^2$ 0.164 and $p < 0.001$).

The Number of Posts impacts the Number of Reactions ($R^2$ 0.163 and $p < 0.001$): That is, for each new post, the Number of Reactions could be increased by 127.26. This result could bring contradictory perspectives regarding the posts' publication frequency and the expected Number of Reactions. Other efforts pointed out that the Number of reactions does not impact users' engagement with the published content [11]. In any case, further research is needed to understand which types of reactions are probably correlated with the types of users' engagement on a LAMs' Facebook, such as user retention as a follower or even user willingness to unfollow the page.

## 5. Discussion

### 5.1. Practical-Managerial Implications

Social media platforms constitute a cost-efficient tool for LAMs to promote their content and services to the society they serve and belong to [12,58]. This same principle is also applied to other domains [59] and not only cultural heritage-related institutions. At the same time, over the last three years, reports in the European context indicate that the need for access to cultural information has increased significantly [60]. Controversially, governmental expenditures for the cultural heritage domain are low, including funding for web presence [61]. In this context, the present study and its proposed methodology can reinforce the cost-efficient use of SMPs to promote LAMs services on the Social Web and, therefore, expand their visibility and awareness. More specifically, through a data-driven approach, a methodological schema has been developed to understand (a) what social media metrics should be included to measure LAMs' Facebook performance, (b) what the current performance of a LAM is through the utilization of descriptive statistics and (c) which are the Administrators actions that bring greater engagement between users and posts. Therefore, the current study could support prior efforts that relied on the SMPs utilization, especially Facebook, as a cost-effective marketing strategy for LAMs [10,12,34]. That is, understanding stakeholders' needs by conducting quantitative survey development [3,10] and then deploying the proposed data-driven methodology to articulate helpful information about users' engagement with the published content through Facebook analytics and metrics.

Apart from verifying SMPs as a cost-efficient strategy for marketing LAMs, the paper also contributes to the administrators' knowledge reinforcement on how to effectively promote services and actions. More precisely, prior efforts indicated the non-sufficient skills of LAMs administrators to use SMPs, even if they believe that these tools are appropriate to promote organizations' services [10,27]. In this sense, the current study contributes two main pillars to LAMs administrators on their knowledge and skills: first, to improve overall web analytics competency while understanding the meaning of the involved metrics and the possible intercorrelations among them [12,62–64]. The study results enable administrators

to develop social media analytics projects within a LAM and, hence, encapsulating and realizing the added value created by the organization's awareness and consideration on the Web [65]. Second, LAMs exhibit a high level of multidisciplinarity, as they bring together scientists and professionals from both humanities and information technologies sectors [66,67]. The study stands as a bridge and supports the efforts to connect the different scientific and professional backgrounds within the LAMs sector. On the one hand, the study overhauls the knowledge of LAMs administrators to use SMPs to improve users' engagement. On the other hand, cultural analytics scientists could adopt the proposed methodology for replication purposes in other organizations apart from the 220 cases.

Lastly, this study enriches benchmarking efforts in LAMs domain through the data gathering method. More specifically, other research efforts utilize benchmarking theoretical lens for improving overall managerial performance [68], remote information services [69] or the quality of the provided services among stakeholders [70]. However, to the best of our knowledge, no prior study provides a benchmarking method to gather a vast amount of LAMs Facebook pages (220 cases) and their performance in terms of administrators' actions and users' engagement. In this respect, administrators could compare their Facebook analytics with data from other LAMs, understand cutting-edge strategies in SMPs management and adopt the best strategies within their organizational context.

### 5.2. Theoretical Implications

One of the study's goals was to provide LAMs with a statistically reliable and validated assessment schema capable of quantifying users' engagement in the Facebook platform through social media analytics and metrics. To implement perform this, we involved several metrics that were categorized into two factors, namely, Administrators Actions and Users' Engagement. This helped practically to understand the cause-and-effect relationship between them. In other words, the study helps social media administrators in understanding which types of posts bring upon greater engagement compared to the others. For example, examining the current dataset of LAMs, video posts returned a higher engagement in all the Users' Engagement metrics compared to the picture or link posts.

Furthermore, different tests took place to prove the statistical reliability and internal consistency of the metrics in each factor. In this way, the study contributes practically to developing a reliable assessment schema that could be utilized in LAMs—or other domains apart from LAMs—while expecting the same reliability level [51] (p. 32) [52,71]. If other researchers repeat experiments in the future to understand how administrators' actions on a Facebook page impact users' engagement, they could adopt this model and its involved metrics as they express statistical reliability and internal cohesion and consistency.

### 5.3. Limitations and Future Work

Developing a statistically significant model to measure users' engagement on SMPs and especially on Facebook opens new research paths to investigate even more LAMs cases compared with the current study. In this sense, we have already started to expand the data sample, including more libraries, archives, museums and their Facebook page analytics and metrics worldwide.

Furthermore, we intend to include other SMPs, apart from Facebook, and hence develop reliable assessment schemes for other platforms. More specifically, based on our research findings (see Figure 4—Average incoming social media traffic share per SMP), we aim to harvest the social media analytics from LAMs on Twitter, YouTube and Instagram. In line with the current methodology, we aim to understand the cause-and-effect relationship between each SMP metrics and provide LAMs administrators with practical suggestions. The ultimate goal is to construct an overall framework of social media analytics and metrics deriving from multiple platforms and how LAMs could benefit from this information.

In conclusion, we encourage related prior studies to integrate the proposed data-driven methodology as a supportive tool in the already established quantitative efforts

by using questionnaires and/or interviews [3,18,27,72]. In this way, LAMs could benefit from the combined use of already established research instruments, while at the same time, valuable conclusions could be drawn about the technological acceptance level of the proposed methodology by the administrators.

**Author Contributions:** Conceptualization, I.C.D., D.K., D.K.-M. and F.G.; methodology, I.C.D., D.K., D.K.-M. and F.G.; formal analysis, I.C.D., D.K., D.K.-M. and F.G.; data curation, I.C.D., D.K., D.K.-M. and F.G.; writing—original draft preparation, I.C.D., D.K., D.K.-M. and F.G.; writing—review and editing I.C.D. and D.K. All authors have read and agreed to the published version of the manuscript.

**Funding:** This research received no external funding.

**Institutional Review Board Statement:** Not applicable.

**Informed Consent Statement:** Not applicable.

**Data Availability Statement:** The datasets used and presented in this study are openly available in Zenodo: Dataset Social Media Analytics and Metrics of Facebook Performance of Libraries, Archives and Museums in https://doi.org/10.5281/zenodo.6361774 (accessed on 11 May 2022) and Dataset Traffic Acquisition to LAMs Websites in https://doi.org/10.5281/zenodo.6505277 (accessed on 11 May 2022).

**Conflicts of Interest:** The authors declare no conflict of interest.

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
