# Peer review of "Social Media Analytics and Metrics for Improving Users Engagement"

_knowledge, doi:10.3390/knowledge2020014_

Round 1
Reviewer 1 Report
The article is engaging, it brings an interesting connection between two topics. Our favorite topics - social media and social media platforms - with the administrations of libraries, archives and museums.
Thanks to the authors for valuable text and interesting topic.
i wish you a lots of success
Please add some ideas about covid and students
P. 1, lines 27-28:
Tkacová, H.; Králik, R.; Tvrdoň, M.; Jenisová, Z.; Martin, J.G. Credibility and Involvement of Social Media in Education—Recommendations for Mitigating the Negative Effects of the Pandemic among High School
Students. Int. J. Environ. Res. Public Health 2022, 19, 2767. https://doi.org/10.3390/ijerph19052767
Tkáčová, H., Pavlíková, M., Tvrdoň, M., & Jenisová, Z. (2021). The Use of Media in the Field of Individual Responsibility for Sustainable Development in Schools: A Proposal for an Approach to Learning about Sustainable Development. Sustainability, 13(8), 4138. https://doi.org/10.3390/su13084138
https://webcache.googleusercontent.com/search?q=cache:25Zotlc_SgkJ:https://www.teof.uni-lj.si/uploads/File/BV/BV2021/01/Tkacova.pdf+&cd=1&hl=cs&ct=clnk&gl=sk
Bogoslovni vestnik/Theological Quarterly 81 (2021) 1, 199—223 Besedilo prejeto/Received:12/2020; sprejeto/Accepted:04/2021 UDK/UDC: 316.7:2 DOI: 10.34291/BV2021/01/Tkacova Hedviga Tkáčová, Martina Pavlíková, Miroslav Tvrdoň and Alexey I. Prokopyev Existence and Prevention of Social Exclusion of Re ligious University Students due to StereotypingAuthor Response
Cover Letter to Reviewer 1
Manuscript entitled: Social Media Analytics and Metrics for Improving Users Engagement
02/05/2022
Dear Reviewer,
We have revised the first version of the manuscript based on your comments and suggestions. We thank you very much for your time and effort in providing valuable remarks that help the manuscript to be improved.
In the following lines, we present in detail our responses to the reviewers' remarks. Moreover, it is noted that the parts that we added to the manuscript are highlighted in yellow colour, while there are minor changes (such as typos) depicted with Track Changes. In this way, changes that have been made to the manuscript are easily tracked.
All authors have approved the revised manuscript and agree to submit it to Knowledge.
Thank you once more for your valuable comments and consideration of this manuscript.
Sincerely,
Ioannis Drivas
Reviewer 1 Comments:
The article is engaging, it brings an interesting connection between two topics. Our favorite topics - social media and social media platforms - with the administrations of libraries, archives and museums.
Thanks to the authors for valuable text and interesting topic.
i wish you a lots of success
Remark: Please add some ideas about covid and students
- 1, lines 27-28:
Tkacová, H.; Králik, R.; Tvrdoň, M.; Jenisová, Z.; Martin, J.G. Credibility and Involvement of Social Media in Education—Recommendations for Mitigating the Negative Effects of the Pandemic among High School
Students. Int. J. Environ. Res. Public Health 2022, 19, 2767. https://doi.org/10.3390/ijerph19052767
Tkáčová, H., Pavlíková, M., Tvrdoň, M., & Jenisová, Z. (2021). The Use of Media in the Field of Individual Responsibility for Sustainable Development in Schools: A Proposal for an Approach to Learning about Sustainable Development. Sustainability, 13(8), 4138. https://doi.org/10.3390/su13084138
https://webcache.googleusercontent.com/search?q=cache:25Zotlc_SgkJ:https://www.teof.uni-lj.si/uploads/File/BV/BV2021/01/Tkacova.pdf+&cd=1&hl=cs&ct=clnk&gl=sk
Bogoslovni vestnik/Theological Quarterly 81 (2021) 1, 199—223 Besedilo prejeto/Received:12/2020; sprejeto/Accepted:04/2021 UDK/UDC: 316.7:2 DOI: 10.34291/BV2021/01/Tkacova Hedviga Tkáčová, Martina Pavlíková, Miroslav Tvrdoň and Alexey I. Prokopyev Existence and Prevention of Social Exclusion of Re ligious University Students due to Stereotyping
Answer: Thank you for that. We add some of the ideas about covid and students. See lines 302-306 (citation n. 45 within the text) (the addition was highlighted in yellow).
Reviewer 2 Report
Social Media analytics and metrics for improving users engagement
The topic is interesting, but the organization of the manuscript could be improved. The title is attractive, but it is not consistent with the content of the manuscript, because readers need a more extensive introduction to the field of social media analytics in relation to user engagement. On the contrary, the manuscript is focused on the case study (LAMs) without placing the readers in the context of social media analytics.
Background
Before addressing the case study, it is necessary to introduce a subsection to deal with user engagement regarding social media. Please see the article “Conceptualising and measuring social media engagement: A systematic literature review” (IJM, 2021).
It is also necessary to review the related literature to connect social media reactions or interaction with user engagement. Please see the article “User reactions to destination brand contents in social media” (ITT, 2016) that uses Facebook Fanpage Karma as a data source. Likewise, it is convenient to review the recent works by Stefania Manca “Digital memory in the post-witness era: How Holocaust museums use social media as new memory ecologies” (2021) and “Exploring tensions in Holocaust museums’ modes of commemoration and interaction on social media” (2022), which also use Fanpage Karma. It would also be interesting to dedicate a subsection to Facebook, reviewing articles that use Facebook Fanpage Reports, and moving lines 237-266 here, to show readers that Facebook is the most appropriate source to address the case study.
Materials and Methods
At the bottom of Figure 4, the source must be indicated.
Conclusions
It is important to highlight the implications. Please insert three subsections in the concluding remarks section such as “Theoretical Implications”, “Practical or Managerial Implications” and “Limitations and Future Work”.
Reference list:
Please check the citations and references (e.g., Abbreviated Journal Name). Multidisciplinary Digital Publishing Institute (MDPI) style. I use Mendeley.com or Zotero.org with the MDPI template.
In Table 1, the reference [10-3] is not clear. Can it be [3-10] or [3, 10]?
Author Response
Cover Letter to Reviewer 2
Manuscript entitled: Social Media Analytics and Metrics for Improving Users Engagement
02/05/2022
Dear Reviewer,
We have revised the first version of the manuscript based on your comments and suggestions. We thank you very much for your time and effort in providing valuable remarks that help the manuscript to be improved.
In the following lines, we present in detail our responses to the reviewers' remarks. Moreover, it is noted that the parts that we added to the manuscript are highlighted in yellow colour, while there are minor changes (such as typos) depicted with Track Changes. In this way, changes that have been made to the manuscript are easily tracked.
All authors have approved the revised manuscript and agree to submit it to Knowledge.
Thank you once more for your valuable comments and consideration of this manuscript.
Sincerely,
Ioannis Drivas
Reviewer 2 Comments:
The topic is interesting, but the organization of the manuscript could be improved. The title is attractive, but it is not consistent with the content of the manuscript, because readers need a more extensive introduction to the field of social media analytics in relation to user engagement. On the contrary, the manuscript is focused on the case study (LAMs) without placing the readers in the context of social media analytics.
Remark: Before addressing the case study, it is necessary to introduce a subsection to deal with user engagement regarding social media. Please see the article "Conceptualising and measuring social media engagement: A systematic literature review" (IJM, 2021).
Answer: Thank you for this valuable remark. Indeed the prior version of the paper lacked what SMAs and social media engagement are from a general point of view. We now added a whole new section improving the paper's logical structure. The new section indicates both what SMAs and social media engagement are, how they are associated, and in which aspects they contribute by improving organizations' web visibility. We also included in the paper the following reference: Trunfio, M., & Rossi, S. (2021). Conceptualising and measuring social media engagement: A systematic literature review. Italian Journal of Marketing, 2021(3), 267-292.
Remark: It is also necessary to review the related literature to connect social media reactions or interaction with user engagement. Please see the article "User reactions to destination brand contents in social media" (ITT, 2016) that uses Facebook Fanpage Karma as a data source.
Answer: We added the proposed reference. In this way, our decision to choose the FanPageKarma tool for data extraction is confirmed by prior investigations that used the same technology. See lines: 285-286.
Remark: Likewise, it is convenient to review the recent works by Stefania Manca "Digital memory in the post-witness era: How Holocaust museums use social media as new memory ecologies" (2021) and "Exploring tensions in Holocaust museums' modes of commemoration and interaction on social media" (2022), which also use Fanpage Karma.
Answer: Thank you for the comment. We had also added the reference of Manca as it is strongly related to the LAMs sector and also uses the same tool for data collection (FanPageKarma). See lines: 285-286.
Remark: It would also be interesting to dedicate a subsection to Facebook, reviewing articles that use Facebook Fanpage Reports, and moving lines 237-266 here, to show readers that Facebook is the most appropriate source to address the case study.
Answer: Using Facebook instead of other SMPs is sufficiently justified based on the following arguments. The first argument is related to the fact that many previous research efforts have also selected the Facebook platform (see lines 291-306). The second and most important argument is related to the results from our preliminary research, which were based on the data collected from the Similar Web API (See lines 307-315). A first-glance analysis of the harvested data indicated that Facebook scores first place in social media traffic generation towards LAMs websites compared to other SMPs. Hence, we believe that these two arguments justify Facebook selection to the paper's readers. Finally, to reflect the above analysis into our paper, adjustments through the methodology section have been made.
Materials and Methods
Remark: At the bottom of Figure 4, the source must be indicated.
Answer: Thank you for that. We revised the figure, and we added the data source of it. We highlighted the lines 309-310 and 601-604.
Conclusions
Remark: It is important to highlight the implications. Please insert three subsections in the concluding remarks section such as "Theoretical Implications", "Practical or Managerial Implications" and "Limitations and Future Work".
Answer: Yes, really, thank you for that. We had previously separated into paragraphs each one of the suggested subsections. Still, as you also mentioned, it will not be so clear to the readers where are the practical, theoretical and future implications. So, following your comment, we separated the Discussion into 3 subsections. First the "5.1 Practical – Managerial Implications", then the "5.2 Theoretical Implications", and finally the "5.3 Limitations and Future Work". In this way, readers will be able to identify each one of the contributions of the study.
Reference list:
Remark: Please check the citations and references (e.g., Abbreviated Journal Name). Multidisciplinary Digital Publishing Institute (MDPI) style. I use Mendeley.com or Zotero.org with the MDPI template.
Answer: We thank you again for this remark. We revised all the citations and added all the abbreviations from the journals.
Remark: In Table 1, the reference [10-3] is not clear. Can it be [3-10] or [3, 10]?
Answer: We revised this point. The right one is 3-10.
Round 2
Reviewer 2 Report
The manuscript has improved significantly.
Good luck!